# Unveiling the Molecular Mechanisms of Glioblastoma through an Integrated Network-Based Approach

**DOI:** 10.3390/biomedicines12102237

**Published:** 2024-10-01

**Authors:** Ali Kaynar, Woonghee Kim, Atakan Burak Ceyhan, Cheng Zhang, Mathias Uhlén, Hasan Turkez, Saeed Shoaie, Adil Mardinoglu

**Affiliations:** 1Centre for Host-Microbiome Interactions, Faculty of Dentistry, Oral and Craniofacial Sciences, King’s College London, London SE1 9RT, UK; ali.kaynar@kcl.ac.uk (A.K.); atakan.ceyhan@kcl.ac.uk (A.B.C.); saeed.shoaie@kcl.ac.uk (S.S.); 2Science for Life Laboratory, KTH-Royal Institute of Technology, 171211 Stockholm, Sweden; woonghee.kim@scilifelab.se (W.K.); cheng.zhang@scilifelab.se (C.Z.); mathias.uhlen@scilifelab.se (M.U.); 3Medical Biology Department, Faculty of Medicine, Atatürk University, Erzurum 25240, Türkiye; hasanturkez@yahoo.com

**Keywords:** glioblastoma, GEMs, co-expression networks, microenvironment, immune response, biomarker, drug target

## Abstract

**Background/Objectives**: Despite current treatments extending the lifespan of Glioblastoma (GBM) patients, the average survival time is around 15–18 months, underscoring the fatality of GBM. This study aims to investigate the impact of sample heterogeneity on gene expression in GBM, identify key metabolic pathways and gene modules, and explore potential therapeutic targets. **Methods**: In this study, we analysed GBM transcriptome data derived from The Cancer Genome Atlas (TCGA) using genome-scale metabolic models (GEMs) and co-expression networks. We examine transcriptome data incorporating tumour purity scores (TPSs), allowing us to assess the impact of sample heterogeneity on gene expression profiles. We analysed the metabolic profile of GBM by generating condition-specific GEMs based on the TPS group. **Results**: Our findings revealed that over 90% of genes showing brain and glioma specificity in RNA expression demonstrate a high positive correlation, underscoring their expression is dominated by glioma cells. Conversely, negatively correlated genes are strongly associated with immune responses, indicating a complex interaction between glioma and immune pathways and non-tumorigenic cell dominance on gene expression. TPS-based metabolic profile analysis was supported by reporter metabolite analysis, highlighting several metabolic pathways, including arachidonic acid, kynurenine and NAD pathway. Through co-expression network analysis, we identified modules that significantly overlap with TPS-correlated genes. Notably, *SOX11* and *GSX1* are upregulated in High TPS, show a high correlation with TPS, and emerged as promising therapeutic targets. Additionally, *NCAM1* exhibits a high centrality score within the co-expression module, which shows a positive correlation with TPS. Moreover, *LILRB4*, an immune-related gene expressed in the brain, showed a negative correlation and upregulated in Low TPS, highlighting the importance of modulating immune responses in the GBM mechanism. **Conclusions**: Our study uncovers sample heterogeneity’s impact on gene expression and the molecular mechanisms driving GBM, and it identifies potential therapeutic targets for developing effective treatments for GBM patients.

## 1. Introduction

Glioblastoma (GBM) is a highly aggressive primary brain tumour that, unfortunately, has a high mortality rate and often recurs despite treatment. The standard therapeutic approach for GBM involves several disciplines, including surgery, radiotherapy, and chemotherapy [1,2]. However, due to the recurrence, the therapeutic options are limited, and the average survival time is around 15–18 months [3]. It is still a high threat to the human population, with a high incidence rate of 3.23 per 100,000 [4]. Widely used chemotherapy drugs for GBM include temozolomide, bevacizumab, and carmustine [5]. However, drug selection may vary for newly diagnosed and recurrent GBM cases. However, some novel treatment strategies are being developed, including immunotherapy, targeted therapies, nanotechnology, gene editing, and hyperthermia, which are still in the early stages of development [6,7,8]. Advanced imaging techniques, such as MRI and DTI, help diagnose GBM and treatment planning by visualising tumour infiltration and aiding surgical decisions [9]. New methods like fluorescence-guided surgery and Laser Interstitial Thermal Therapy (LITT) improve tumour removal precision. Emerging therapies like PROTACs target specific proteins to disrupt tumour growth, offering personalised treatment options [10].

GBM presents a profound challenge due to its heterogeneity and the complex interaction with its microenvironment. The interplay between tumour cells and the tumour microenvironment cells is key in tumour initiation, invasion, and resistance to therapy. GBM cells interact with non-neoplastic cells such as tumour-associated macrophages (TAMs), microglia, T lymphocytes, astrocytes, endothelial cells, and pericytes, contributing to various tumour niches—angiogenic, invasive, and hypoxic [11,12,13]. Particularly, microglia play a key role by secreting pro-inflammatory mediators like nitric oxide, TNF-α, and interleukins, fostering neuroinflammation and compromising the blood–brain barrier [14]. The prognosis of GBM is influenced by its biological pathways and gene expression profiles, implicating important cellular and molecular interactions within its milieu [15,16,17].

GBMs are classified into primary, often arising de novo in adults and requiring IDH-wildtype, which is associated with shorter survival, showing 15 months overall survival and 7–9 months progression-free survival, and secondary, developing from lower-grade astrocytoma and requiring IDH-mutant, which is better prognosis showing 31–46 months overall survival and 11–20 months progression-free survival [18,19,20]. This distinction is critical for diagnostic and prognostic implications, where genes such as *IDH1*/*IDH2*, *MGMT* promoter, *EGFR*, *TP53*, and *PTEN* play a significant role [17,21,22]. Consequently, the WHO categorises GBM as grade IV glioma, characterised by its malignant nature, active mitosis, and a tendency towards necrosis. Furthermore, GBM can be stratified into molecular subtypes, including classical, mesenchymal, and proneural, based on genetic alterations and miRNA expression profiles, which reveal five clinical subtypes: astrocytic, neural, oligoneural, radial glial, and neuromesenchymal [16,23]. 

Tumour formation resulted in alteration in the functioning of several biological systems. Cancer is a highly proliferative disease with complex molecular mechanisms underlying it, but it is also considered a metabolic disease in which some major metabolic shifts also occur [24]. Growing tumours rearrange their metabolic programs to meet the bioenergetic and biosynthetic demands of sustained cell growth [24,25,26]. The metabolic profile in glioma cells is exemplified by increased glucose and glutamine consumption, activation of metabolic enzyme isoforms, increased lactate production and secretion, and a metabolic shift toward aerobic glycolysis with reprogramming of energy metabolism known as the Warburg effect [27].

Systems biology is an interdisciplinary approach that combines experimental and computational methods to understand complex biological systems as a whole [28]. It offers a comprehensive view by employing high-throughput omics data, including transcriptomics, and by utilising statistical and network-based analyses [28,29]. Genome-scale metabolic models (GEMs) are the collection of biochemical reactions and associated enzymes and transporters, which represent the cell mathematically [5,30]. The application of GEMs not only enhances our understanding of the biological complexities of GBM but also helps to discover novel therapeutic strategies specific to individuals, which may revolutionise GBM treatment [30,31]. 

In this study, the major aim was to utilise an integrated network approach, incorporating condition-specific GEMs and co-expression network analysis, to elucidate the molecular mechanism of GBM. Our focus was particularly on identifying potential therapeutic targets and biomarkers. In conjunction with these investigations, the impact of tumour purity scores (TPSs) on gene expression patterns and their interactions within the tumour microenvironment (TME) was assessed. This assessment assumes that TPSs may reflect the origins of transcripts within the sample, suggesting that a higher TPS indicates a predominance of RNA originating from tumour cells, while a lower TPS points to significant contributions from non-tumour cells in the surrounding environment, which has the potential to reveal significant transcriptomic variance associated with TPS and to illuminate the gene expression patterns accordingly. This analysis was designed to reveal the complex interplay influenced by sample heterogeneity, guiding our approach to identifying effective targets. These objectives were pursued through a structured workflow utilising RNA-seq data from The Cancer Genome Atlas (TCGA), as detailed in Figure 1. We conducted a two-way analysis focusing on metabolic alterations and gene-to-gene relationships. By generating GEMs specifically tailored to each GBM patient subgroup defined by the TPS, we provided insights into the complexity of GBM through information derived from the impact of sample heterogeneity. Through genome-scale metabolic network analysis, reporter metabolite analysis, and gene-centric investigation, genes that could be considered potential biomarkers and therapeutic targets were identified. The co-expression network analysis was also performed to examine the genes identified through their co-expression with neighbouring genes that may indicate the gene regulatory property of GBM. These perspectives assist the development of personalised and effective treatment strategies aimed at improving patient outcomes and addressing the challenges of this cancer, as depicted in our study workflow (Figure 1).

## 2. Materials and Methods

### 2.1. Data Collection and Processing

We obtained the RNA-seq dataset for the TCGA GBM project from the Genomic Data Commons (GDC) platform [32]. This dataset comprises mRNA expression levels as Transcript Per Million (TPM) and STAR read count, along with clinical data for 174 tumour samples and five control samples from normal tissues (NT). We excluded duplicate samples and included samples that have tumour purity score (TPS) based on clinical information, which resulted in a total of 144 primary solid tumour samples with well-documented clinical details, including survival time and the last follow-up recording. The protein-coding genes based on the dataset’s inherent annotation as “protein_coding” were selected. The data retrieval and processing were performed using the TCGAbiolinks R package (version 2.32.0) within the RStudio platform (version 2024.4.2.764) [33].

Later in the study, raw count data were normalised by using DESeq2 [34]. Following that, the genes with low expression based on the mean of count values lower than 10 value were removed. Tumour samples were divided into three subgroups according to TPS: High TPS, Low TPS, and TP (including 144 tumour samples).

### 2.2. Transcriptome Analysis

The study utilised gene count values from selected samples in the DEG analysis after removing low-expressed genes. The samples were sorted according to the TPS and grouped into High and Low TPS, with the first quantiles (25) and third quantiles (75), respectively. The DEG analysis was conducted using the R package “DESeq2” (version 1.44.0) operating on the R platform (version R version is 4.4.0) [33]. The study findings contribute to a better understanding of the gene expression patterns associated with TPS in GBM, which may have implications for treatment decisions and patient outcomes. We compared subgroups for gene set enrichment analysis by selecting certain cut-off values as 1 × 10^−10^ to define significantly altered genes. The GO term lists from Ensembl Biomart were retrieved, which were accessed from Ensembl using the “getBM” function [35]. The functional enrichment analysis of the DEGs was carried out using the “enrichGO” function from the “clusterProfiler” R package (version 4.12.0) [36]. GO terms that have Benjamini and Hochberg adjusted *p*-value of lower than 0.05 were considered statistically significant. For pathway enrichment analysis of DEGs, we used the “enrichKEGG” function from the “clusterProfiler” R package to obtain the KEGG pathway terms. KEGG pathway terms with a Benjamini and Hochberg adjusted *p*-value lower than 0.05 were considered statistically significant.

### 2.3. Genome-Scale Metabolic Modelling

The Human GEM 1 (version 1.14.0) metabolic model [37], which includes reaction information available in the literature, was used as the reference model to generate GEMs specific to patient groups. The arithmetic mean of gene expression levels was used as input for the reconstruction of condition-specific GEMs. RAVEN Toolbox (version 2.8.0) [38] and the tINIT algorithm [39] were used to generate condition-specific GBM models. The simulations were performed on the MATLAB platform (R2023a version 9.14.0.2206163) [40]. Each model underwent testing to determine their proficiency in accomplishing the task designed for the HUMAN GEM 1, and generated models demonstrated successful performance on the test, which suggests these models possessed the required capabilities for further analysis. 

Additionally, the created models were compared structurally. This comparison was made using the compareMultipleModels function, which creates a binary matrix and using an essential task list that indicates the presence or absence of reactions in each model and subsystems of each model to show the similarities and differences between models. Another comparison facilitated by the compareMultipleModels function provides a comprehensive understanding of how reactions are distributed across models, highlights relative changes in sub-coverage, and identifies variations that highlight differences or similarities in subsystems between the models. This comparison enables us to gain critical insights into the functional characteristics of the models. Flux Balanced Analysis (FBA) was performed with each patient-specific GEM using the solveLP function from RAVEN Toolbox, with reported constraints, Ham’s media, and defining the Biomass reaction (MAR13082) as the objective function.

### 2.4. Reporter Metabolite Analysis Using GEMs

GBM-specific metabolic models were generated based on the tumour purity score (TPS), resulting in three distinct models: TP, High TPS, and Low TPS. An additionally model was generated from NT data: the NT model. Subsequently, to elucidate the altered metabolic profiles, we conducted a reporter metabolite analysis by integrating the results of DEG analysis with the GBM metabolic models tailored to each patient group. Reporter metabolite analysis was performed using the reporterMetabolites function from RAVEN Toolbox using GEMs specific for each patient group. Three categories of potential reporter metabolites were identified, encompassing upregulated gene-related metabolites, downregulated gene-related metabolites, and a combined list. The results were compared across the four models, revealing metabolites that significantly changed across groups. Furthermore, the TPS-specific metabolites were examined to elucidate the effect of the tumour microenvironment.

### 2.5. Gene Essentiality Analysis Using GEMs

In gene essentiality analysis, the generated models, the NT model, TP model, High TPS model, and Low TPS model, were used. This analysis involved the simulation of single gene deletions for each model, utilising the COBRA toolbox and RAVEN toolbox, enabling us to systematically evaluate the significance of individual genes within the context of GBM metabolism. We used the checkTasksGenes function from the RAVEN toolbox, which utilises essential tasks, and the singleGeneDeletion function from the COBRA toolbox. To ensure biological relevance, HAM media composition was imposed as a constraint on the models. Biomass reaction was selected as the objective function, which reflects cellular growth and proliferation. This comprehensive approach allowed us to uncover essential genes specific to each group, potentially revealing personalised therapeutic targets for GBM treatment.

### 2.6. Survival Analysis and TPS Correlation 

Our study utilised univariate Cox regression models and Kaplan–Meier survival analysis to investigate the relationship between gene expression levels and survival rates in GBM patients. Hazard ratios for each gene were computed, categorising them based on their impact on survival, with genes associated with a *p*-value less than 0.05 classified as significant. Genes linked to poorer survival outcomes were identified as unfavourable prognostic genes, while those associated with better outcomes were termed favourable prognostic genes. Statistical analyses were conducted using the R package “survival” (version 3.7.0). To further elucidate the interaction of gene expressions with the tumour microenvironment, we performed spearman correlation analyses between each gene’s expression level (TPM value) and the tumour purity score (TPS). This enabled the identifications of both negative correlations, which may indicate transcript source dominated by non-tumorigenic cells within the sample, and positive correlations, suggesting higher dominance of tumour cells. This comprehensive approach highlights prognostic markers and enhances our understanding of the molecular and environmental dynamics of GBM, ultimately guiding more tailored and effective therapeutic strategies. 

### 2.7. Gene Co-Expression Network Analysis

For co-expression analysis, we utilised RNA-Seq data from TCGA to explore gene expression profiles related to the GBM mechanism. Gene expression data (tpm_unstrand) was retrieved from the TCGA dataset, in which samples have TPS. The expression data was converted to a numeric matrix, and genes with an average TPM of less than 1 were filtered out to focus on genes with significant expression levels. A correlation matrix was constructed using spearman’s rank correlation coefficient to measure co-expression between genes, including only the top 5% of gene pairs with the highest correlation values (|R| ≥ 0.95). A network graph was generated from the correlation matrix, where nodes represent genes and edges represent significant co-expression relationships. Interactive modules were identified within the network using a random walk method. Modules with more than 30 nodes and connectivity (transitivity) greater than 0.5 were labelled as “High Connectivity Clusters (HighCC)”, while others were labelled as “Low Connectivity Clusters (LowCC)”. The identified modules were annotated and prepared for visualisation in Cytoscape [41], with node and edge information saved for further network analysis.

TPS-correlated gene expression patterns were analysed within the identified modules. Overlap analysis was performed using the hypergeometric test to determine the enrichment of marker genes within each module. Network features such as degree, betweenness, and closeness centrality were calculated for each node within the identified modules to understand their topological importance. This analysis was repeated for multiple significant modules to capture the network dynamics comprehensively. The entire workflow was implemented in R, leveraging packages such as igraph (version 2.0.3). 

## 3. Results

### 3.1. Gene-Set Enrichment Analysis

DEG analysis was conducted using significantly altered genes across different determined groups, applying a cut-off value of 1 × 10^−10^ to identify significantly altered genes (Appendix A). NT and TP comparison revealed 1174 differentially expressed genes, comprising 510 upregulated and 664 downregulated genes. Additionally, the NT vs. High TPS comparison identified 1509 differential expressed genes, including 598 upregulated and 911 downregulated genes. The comparison of NT and Low TPS demonstrated alterations in 1162 genes, with 803 upregulated and 359 downregulated. Finally, the High TPS vs. Low TPS comparison showed 917 significantly altered genes, with 184 upregulated and 733 downregulated. Differentially expressed genes provide distinct profiles across the TPS groups. Many of the biological processes identified through the Gene Enrichment Analysis of NT vs. TP are closely tied to cell division and regulation, key factors in the underlying mechanisms of GBM (Figure 2). These processes could encounter significant alterations that result in uncontrolled cell proliferation, a defining characteristic of cancer. Notably, glioma cells exhibit abnormal cell division, dysregulation of cell cycle processes and chromosomal instability, which emphasises the significance of these process alterations and contributes to the unrestrained growth of glioma cells [42]. Examples of genes involved in significantly altered biological processes include *BIRC5* (anti-apoptotic), *CDK1* and *CDK2* (cell cycle regulation) (Appendix A), *TP53* (tumour suppression), and *PTTG1* (mitosis regulation), highlighting key mechanisms driving GBM cell proliferation.

When discussing TPS in the context of GBM, High TPS indicates a higher proportion of tumour cells within the sample, whereas Low TPS indicates a relatively higher presence of non-tumour cells, such as immune cells. Differences in biological processes and/or KEGG pathways in the High and Low TPS subgroups may provide insights into tumour biology, TME, and immune responses.

Considering Low TPS, enriched genes involved in immune responses or non-tumorigenic cell activity might be expected. This could include pathways related to leukocyte activation, cytokine production, and other immune system processes. These gene sets associated with lower TPS indicate a more distinct immune microenvironment with the active participation of immune cells. In line with this projection, significantly altered biological processes, particularly regulation of innate and adaptive immune responses, are positive regulation of cytokine production, immune response-regulating signalling pathway, leukocyte mediated immunity, and T cell activation (Figure 3B). The activation of immune and infection-related pathways illustrates the complex interactions between the GBM cells and the TME. Some of the significantly altered KEGG pathways (Figure 3D) are phagosome, antigen processing and presentation, complement and coagulation cascades, and natural killer cell-mediated cytotoxicity. Low TPS results revealed significantly changed pathways involving tumour–non-cancerous cell interactions, such as PD-L1/PD-1 (*CD274* on tumour cells and *PDCD1* on T cells) and Galectin-9/TIM-3 (*LGALS9* on tumour cells and *HAVCR2* on immune cells). These protein interactions are manipulated by GBM cells to suppress T-cell activity, evade immune detection, and foster tumour growth and survival.

On the other hand, the High TPS group may show enrichment of gene sets that reflect the intrinsic properties of the tumour cells, which could be associated with cell proliferation. This indicates that the tumour cells are in active growth and metabolic reprogramming state, indicating aggressive tumour behaviour. In line with these, some significantly altered biological processes come forward, such as DNA replication, mitotic nuclear division, vesicle-mediated transport in the synapse, neurotransmitter secretion, presynaptic endocytosis, and regulation of chromosome separation (Figure 3A). Several KEGG (Figure 3C) pathways significantly altered that influence tumour progression. The cell cycle and DNA replication pathways, which are affected by cyclins and CDKs, are critical. Dysregulation of them could lead to cell division and genomic instability [42]. The phosphatidylinositol signalling through the PI3K/Akt pathway [43] and the calcium signalling pathway [44] both significantly affect cell survival and proliferation. Targeting these pathways could disrupt survival signals and impair tumour cell functioning. Moreover, the cAMP signalling pathway regulates GBM’s response to external signals, which could promote growth or resistance to cell death [45], which is another therapeutic option that alters cAMP levels. Lastly, the cellular senescence pathway, usually protective, is often inactivated in GBM, allowing uncontrolled growth. Reactivation of this pathway may terminate GBM by inducing growth arrest. Together, these pathways highlight the intricate cellular mechanisms of GBM, which presents multiple options for targeted therapy.

### 3.2. Structural Comparison of Generated Metabolic Models

GBM is an aggressive and malignant brain cancer characterised by uncontrolled cell growth and poor prognosis. Understanding the underlying metabolic alterations in GBM is crucial for the development of effective therapeutic strategies. In this study, we explored the structural comparison of generated metabolic models (NT, Low TPS, High TPS, and TP) (Figure 4A) and identified the links between significantly different metabolic processes and GBM.

Significantly altered metabolic genes, including *GPX1*, *GPX7*, and *GPX8*, which are crucial for oxidative stress response by scavenging free radicals, helping to prevent lipid peroxidation and maintaining intracellular homeostasis and the redox balance. Additionally, Heme oxygenase-1 (*HMOX1*) breaks down heme into iron and other by-products, indicating potential toxicity effects within the TME (Appendix A). *FLAD1*, which shows favourable prognostic features along with positive correlation with TPS, is essential for vitamin B2 (riboflavin) metabolism, facilitating mitochondrial function and the high energy demands of tumour cells energy demands. Conversely, *ACP3*, *ACP5*, and BLVRB are elevated in the Low TPS, with *ACP3* modulating immune responses and impacting the microenvironment (Appendix A). *ACP5* and *ACP6* are unfavourable prognostic genes to tumour growth and immune system evasion, highlighting their role in cancer progression. *FUT1*, *FUT2*, *B3GNT2*, *GCNT2*, and *ST8SIA5* are involved in the blood group biosynthesis, a synthesis of blood group antigens process, which are glycan structures on cell surfaces that can influence tumour cell interactions with the TME, including immune system components (Appendix A). Metabolic genes *CYP1B1*, *AANAT*, *ADH5*, and *ALDH3A2*, associated with the serotonin and melatonin biosynthesis pathway, contribute to tumour growth, immune modulation, and altered metabolic states in the TME (Appendix A). Particularly, *CYP1B1* is a critical metabolic enzyme for melatonin metabolism, including the hydroxylation of melatonin. *AANAT* is required to convert serotonin to N-acetylserotonin, a precursor in the melatonin synthesis pathway. 

Structural comparison of metabolic models (NT, High TPS, Low TPS, and TP) has revealed intriguing differences in five essential metabolic tasks, which may shed light on possible links to GBM (Figure 4B).

De novo synthesis (minimal substrates with AA, minimal excretion) of tyrosine from basic building blocks such as amino acids, while minimising the need for additional substances and reducing waste products, appears to be minimal or absent in GBM tumour models, which could be indicative of necessities of tyrosine metabolism in GBM. Due to the decreased de novo synthesis of tyrosine, the cells might utilise the amino acids from the existing pool or uptake tyrosine from the external environment. This reliance on available amino acids, which should be derived from other sources or extracellular sources, may make the tumour cell more reliant on external supplies of tyrosine to meet its metabolic demands.

NAD+ and NADH are key molecules in cellular metabolism and energy production. They act as co-enzymes in different biological reactions [46]. The de novo synthesis of NAD+ and NADP (minimal substrates, physiological excretion) is crucial for redox reactions and energy metabolism. However, the absence of this synthesis pathway in Low TPS and TP models of GBM suggests a potential dependence on salvage pathways to maintain their NAD+ and NADP needs. On the other hand, this pathway is active in the High TPS model. This alteration in salvage pathways indicates specific metabolic adaptations associated with dysregulated NAD+ and NADP metabolism, potentially impacting tumour progression. The lack of de novo synthesis of these molecules might make the tumour cell more vulnerable. For example, therapies that limit these molecules’ availability could theoretically inhibit the tumour cells’ growth and proliferation. The Low TPS model indicates that *HPRT1* and *TYMP* are crucial in the ATP salvage pathway (Appendix A). *HPRT1* plays an important role in the recycling of nutrient-scarce purines in the hypoxic TME, while the role of *TYMP* in pyrimidine metabolism and angiogenesis suggests its contribution to tumour proliferation.

Variations in essential metabolic functions may reflect intrinsic differences in energy generation, biosynthesis, and oxidative stress responses in specific GBM models. These insights offer potential pathways for developing targeted therapies tailored to the distinct metabolic needs of individual GBM cases, to enhance patient outcomes and manage this aggressive disease more effectively. Further studies are required to explore these metabolic associations and their impact on the pathogenesis of GBM extensively. 

### 3.3. Reporter Metabolite Analysis

GBM-specific metabolic models were generated by considering the tumour purity scores (TPSs), High TPS model, Low TPS model, TP model, and the control model (NT). To reveal the altered metabolic profile, we performed a reporter metabolite analysis using the DGE analysis result (NT vs. TP and High TPS, Low TPS) with GBM metabolic models specific to each GBM patient group. Three potential reporter metabolites lists, upregulated-gene-affected (URGA) metabolites, downregulated-gene-affected (DRGA) metabolites, and a total list that includes both were obtained for each model provided in Appendix A. 

Reporter metabolites from the comparison between the NT model and the TP model provide valuable insights into the potential metabolites associated with GBM and its underlying mechanisms. The prominent URGA metabolites and their biological relationships with GBM were investigated (Table 1).

We visualised URGA metabolites extensively in Appendix A. Several key metabolites and metabolic pathways were affected differently between the High TPS and Low TPS subgroups. In the Low TPS group, arachidonic acid metabolism components were identified as featured reporter metabolites. Metabolites from this pathway, including hepoxilin A3 and leukotrienes, are crucial for immune response modulation. Hepoxilin A3, derived from arachidonic acid, promotes a neutrophil-based inflammatory response. Leukotriene A4, primarily synthesised by leukocytes, is a source of leukotriene B4, which stimulates inflammatory cells, and leukotriene E4, which is involved in eosinophil recruitment and vascular effects. These significant metabolites indicate their vital role in modulating immune responses within the tumour microenvironment. Moreover, chondroitin and keratan sulfate, essential for extracellular matrix organisation and cell signalling, were also significantly altered in the Low TPS group. This may suggest their involvement in creating a supportive environment for tumour progression.

On the other hand, the High TPS group showed a distinct metabolic profile. Folate metabolism, including tetrahydrofolate and dihydrofolate, which are important for nucleotide biosynthesis and repair mechanisms, are significantly altered. This supported the rapid proliferation of GBM cells by ensuring an adequate supply of nucleotides for DNA replication and repair. Additionally, putrescine, a polyamine involved in cellular proliferation and differentiation, is significantly altered, which may indicate aggressive metabolic reprogramming in the tumorigenic group. Moreover, the fatty acid and carnitine shuttle pathways were significantly altered in the High TPS group, which could support the energy demands and biosynthetic processes of rapidly proliferating GBM cells.

These URGA metabolites are associated with several critical metabolic processes that are relevant to immune response, matrix organisation, cancer cell growth, invasion, and survival. The dysregulation of these processes may contribute to the development and progression of GBM. Further research into the specific roles of these metabolites in GBM mechanisms could provide valuable insights for potential therapeutic strategies targeting these metabolic pathways.

The prominent DRGA metabolites and their biological relationships with GBM were investigated (Table 2). The DRGA metabolites in the comparison of the NT model and TP model provide valuable insights into the metabolic alterations associated with GBM.

Given that these DRGA metabolites are significantly associated with processes critical for cancer cell survival and proliferation, they may help understand the mechanism of GBM. GBM cells appear to exhibit altered amino acid metabolism, relying on alternative energy sources, such as ketones. Furthermore, the dysregulation of lipids involved in signalling pathways suggests that GBM cells may manipulate signalling cascades to support their growth and avoid mechanisms that would normally limit their proliferation or cause cell death. Glutamine addiction in GBM is a metabolic shift that leads to the conversion of glutamine to glutamate and then to alpha-ketoglutarate (AKG). In particular, the alteration of glutamate’s metabolic activity in lysosomes, rather than contributing to mitochondrial energy production, may indicate a shift toward signalling and stress response functions. Understanding these metabolic changes is crucial for developing targeted therapies against GBM and shedding light on the complex mechanisms of the disease.

### 3.4. Gene Essentiality Analysis

The single gene deletion results (Appendix A) reveal the functional significance of various genes, which are specific to GBM models (Figure 5), and their roles in biological processes. In silico deletion of the *SLC27A5* gene (specific to the TP model), which encodes a solute carrier involved in hepatic fatty acid uptake and bile acid reconjugation, could disrupt crucial metabolic processes [55]. Deoxycytidine kinase (*DCK*: specific to the Low TPS model) plays a pivotal role in phosphorylating deoxyribonucleosides and nucleoside analogues, which makes it a critical factor in the body’s response to antiviral and anticancer chemotherapeutic agents. *CMPK1* (Cytidine/uridine monophosphate kinase 1) (specific to the Low TPS model), associated with nucleic acid biosynthesis and pyrimidine nucleotide de novo synthesis, underscores the importance of maintaining balanced nucleotide metabolism [56].

The *QPRT* gene is included in neuronal health by breaking down quinolinate [57], which could otherwise harm neurons and is associated with neurodegenerative diseases. Moreover, the *KYNU* gene encodes kynureninase, an enzyme essential for the biosynthesis of NAD from tryptophan [58]. *CYP27A1* is another crucial gene that has a role in bile acid metabolism, cholesterol metabolism, steroid synthesis, and lipid biosynthesis, further emphasising its relevance in regulating cholesterol levels and preventing rare lipid storage diseases [59,60,61].

*HAAO* plays a pivotal role in central nervous system function. The enzymatic activity of *HAAO* in converting 3-hydroxy anthranilic acid to quinolinic acid could influence neurological and inflammatory systems. The GUSB gene encodes a hydrolase crucial for glycosaminoglycan degradation. Mutations in this gene could lead to mucopolysaccharidosis type VII disease (MPS VII), which indicates its role in lysosomal function and metabolic disorders [62].

*NADSYN1* is a critical gene involved in NAD+ biosynthesis [63], a co-enzyme essential for metabolic redox reactions, cell signalling, and protein post-translational modifications. The final step in NAD+ biosynthesis is catalysed by NAD synthetase, highlighting the importance of this gene in maintaining the delicate balance of cellular metabolism.

Deletion of these genes can have profound implications for various metabolic, cellular, and physiological processes, shedding light on their roles in health and disease. Understanding the functions and significance of these genes contributes to our knowledge of biology and provides potential targets for therapeutic interventions.

The pathways shown in Figure 6 highlight the key role of tryptophan metabolism in the synthesis of NAD, which is vital for energy production and cellular health. The process begins with tryptophan and progresses through several reactions facilitated by enzymes such as *KYNU*, *HAAO*, *QPRT*, *NADSYN1*, and *NMNAT1*. These enzymes were predicted to be critical in cancer models through gene essentiality analyses and offer potential targets for cancer therapy by disrupting cancer cell metabolism.

The figure expresses the transport protein *SLC7A7*, which shows a negative correlation with TPS, which indicates its potential regulatory role in cellular uptake processes that may impact both kynurenine and polyamine pathways along with several amino acid transports. The interplay between kynurenine and polyamine metabolism, which may influence immune modulation, is also depicted. The activation of the aryl hydrocarbon receptor (*AHR*) by kynurenines, which enhances polyamine production, establishes a regulatory loop potentially relevant to cancer’s immunological aspects. The positive correlation of enzymes such as *ODC1* and *SRM* in the polyamine pathway underscores the metabolic imbalances often associated with cancer, emphasising the complex network of metabolic interactions that can influence tumour progression and immune responses. The Warburg effect is well known for its metabolic reprogramming, known as aerobic glycolysis, which affects other metabolic pathways, such as the kynurenine pathway, which is part of tryptophan metabolism. Kynurenine acts as an immunosuppressive messenger metabolite that promotes the expansion of regulatory T cells (Tregs) and suppresses effector T cell activity, thus contributing to the characteristic immunosuppressive environment of GBM [64].

Furthermore, we examined two combined lists of essential genes from four models derived from both the essential metabolic task genes and single gene deletion studies to determine which pathways are essential. We examine those gene lists on the STRING database to get gene networks [65], which highlight processes of tRNA aminoacylation for protein translation, aminoacyl-tRNA ligase activity, and cholesterol biosynthesis (Appendix A). The single gene deletion approach predominantly affects nuclear pore organisation, mitochondrial electron transport from cytochrome c to oxygen, quinolinate metabolic processes, de novo NAD biosynthetic processes, transcription by RNA polymerase, and retinoic acid metabolism.

### 3.5. Survival Analysis and Correlation Analysis

In this research, we utilised Kaplan–Meier (KM) and Cox regression (Cox) analyses to explore the association between gene expression levels and patient survival outcomes in the TCGA cohort. Our findings identified 7960 and 2232 prognostic genes from KM and Cox analysis, respectively, of which 2043 of them are common (Appendix A). Additionally, we conducted Spearman correlation analyses to assess the relationship between gene expression and tumour purity score (TPS). Based on the results, genes were categorised by their correlation strength; those with high correlation had a coefficient of abs(0.5) or above, while medium correlation was defined as coefficients between abs(0.5) and abs(0.3), covering both positive and negative correlations. From this, 1737 genes demonstrated a high correlation, and 4418 genes showed a medium correlation (Appendix A). Our analysis primarily focused on genes with high correlation to derive significant insights. 

### 3.6. Co-Expression Analysis Reveals Significant Modules Relevant to the Correlated Genes

In this study, we identified a total of 32 modules based on gene expression profiles. Out of the 32 modules identified, 25 demonstrated high connectivity (HighCC) properties. Notably, modules 7, 8, and 36 showed significant alignment with genes that are positively correlated (R ≥ 0.5) according to the hypergeometric test results (Appendix A). In contrast, modules 67 and 13 matched significantly with negatively correlated genes (R ≤ −0.5).

This differentiation in modules based on their correlation with gene expression highlights the intricate network dynamics and potential key players in GBM pathogenesis. The positively correlated modules likely represent genes that are co-expressed and potentially co-regulated, contributing to tumour growth and progression. Conversely, the negatively correlated modules may include genes that are involved in immune responses. These findings provide a foundation for further investigation into the molecular mechanisms underlying GBM and identify potential targets for therapeutic intervention.

### 3.7. Discovery of Potential Genes for the Development of GBM Therapy

To identify therapeutic targets for GBM, we conducted a multifaceted analytical approach by integrating differential gene expression (DEG) analysis, correlation analysis, and tissue expression and glioma expression profile from the Human Protein Atlas [66,67]. The significantly correlated genes of 265 out of 290 glioma genes with the TPS score supports the efficacy of our approach in identifying cancer-related genes linked to both TPS groups. The correlation analysis was particularly useful in recognising gene expression influenced by non-tumorigenic cells in the Low TPS group and dominated by tumour cells in the High TPS group. This distinction enabled us to decipher the microenvironment’s impact, as well as intrinsic tumour effects.

*GSX1* and *SOX11* (Figure 7, Figure 8 and Figure 9) have been identified as critical marker genes for the High TPS group, indicating their significant influence on tumour dynamics. *GSX1* plays a critical role in regulating neural progenitor cells, consequently aiding in the maintenance of cellular stemness and differentiation [68], which enhances the complexity and adaptability of cancer cells. A study shows that the silencing of *GSX2* in pancreatic cancer cells resulted in a decrease in their proliferation, migration, and invasion, along with an increase in apoptosis and enhanced sensitivity to gemcitabine treatment [69]. Both *GSX1* and *GSX2*, as homeobox genes, are involved in neural development through their regulation of the differentiation and maintenance of specific neural cell types.

*SOX11* is critical for neurogenesis and neuronal differentiation and also significantly contributes to cellular survival, promoting tumour cell proliferation, invasion, and resistance to apoptosis [70], which facilitates tumour malignancy and progression. *SOX11* has been identified with a high positive correlation with the tumor purity score (TPS), which is particularly upregulated in High TPS. Moreover, *SOX11* demonstrates a notably higher expression in brain cancer compared with other cancer types, underscoring its specific involvement in glioma pathology [67] and its potential as a target for therapeutic interventions in this cancer subtype. Moreover, *SOX11* shows significantly higher expression in both the mutant IDH and the mutant ATRX groups, suggesting its involvement in metabolic pathways altered by IDH mutations and its role in chromatin remodelling and genomic stability influenced by ATRX mutations.

The “RNA cancer specific FPKM: Glioma” filter may exclude marker genes related to non-cancerous cells. Therefore, to identify immune response-related marker genes, those with a high negative correlation out of the 290 glioma genes were also considered. This approach helped construct a comprehensive list of Low TPS-associated marker genes. Concurrently, several key genes negatively correlated with TPS and expressed in the brain, including *TREM2* and *LILRB4*, have emerged. The immune-related functions of *LILRB4*, coupled with its increasing relevance in tumour immunology, warrant further investigation into its role in tumour-immune dynamics and its potential for therapeutic targeting.

*LILRB4* (Figure 7, Figure 8 and Figure 9), a member of the leukocyte immunoglobulin-like receptor (LIR) family, plays an essential role in the immune system. It is expressed on monocytic cells and transmits a negative signal that inhibits the activation of the immune response. This receptor is involved in antigen capture and presentation, managing inflammatory responses and cytotoxic activities to fine-tune the immune response and minimise autoimmunity. The role of *LILRB4* in interacting with tumour-associated macrophages and its negative regulation of immune responses in tumours highlights its potential as a target for immunotherapy [71], underlining its pivotal role in modulating tumour-immune interactions. 

Polyamine metabolism, kynurenine metabolism, and de novo NAD metabolism are also connected to this marker network (Figure 8). Several genes from the marker network have been studied. Transcription factors (TFs) like *ASCL1* and *OLIG2*, which are co-expressed in GBMs (Module 8), play crucial roles in maintaining tumour cell heterogeneity and hierarchy. Dysregulation of these TFs by oncogenic mutations results in tumour development and progression. *ASCL1* and *OLIG2* dynamically interact, affecting tumour cell types, migration, and proliferation. Notably, high *ASCL1* levels are linked to glioma stem cells (GSCs), which are characterised by increased proliferation, migration, and therapy resistance [72,73].

These TFs not only sustain tumour cell maintenance but also contribute to therapy resistance and tumour recurrence. Brain tumour-initiating cells (BTICs), akin to GSCs, are difficult therapeutic targets located within the blood–brain barrier. Genetic and epigenetic research has enabled novel therapeutic approaches, such as RNAi-mediated targeting of transcription factors (e.g., *SOX2*, *OLIG2*, *SALL2*, *POU3F2*) [74]. These strategies show the potential to prolong survival in preclinical models, particularly when applied via methods such as convection-assisted delivery.

Our analysis revealed that *NCAM1* (Neural cell adhesion molecule 1) is one of the top three central genes in module 8, showing a positive correlation and brain-enhanced expression according to the Human Protein Atlas. This gene is prominently featured in Figure 8, underscoring its centrality and potential importance in GBM pathology. *NCAM1*’s roles in neurogenesis and angiogenesis are particularly relevant in GBM, where the tumour microenvironment requires vascular support and neural integration to sustain its growth and invasiveness [75]. Targeting *NCAM1* and its associated pathways could offer novel therapeutic strategies for managing GBM. Its centrality in the co-expression network and positive correlation with GBM pathology also underscores its importance.

Our analyses revealed significant findings, particularly when high correlation and unfavourable genes were filtered. Notably, most of the genes are downregulated, especially in High TPS, which suggests profound biological and metabolic transition within the tumour. *MAP3K1* emerged as a key gene; it was upregulated, an unfavourable prognostic gene according to both Kaplan–Meier and Cox analysis, and showed a positive correlation, suggesting that it was directly influenced by tumour cells. Conversely, in Low TPS, several genes, such as *ALOX5*, were identified as negatively correlated, upregulated in DEG, and unfavourable prognostic genes according to both Kaplan–Meier and Cox analyses. *ALOX5*, in particular, was significant for its clear role in immune responses, further substantiated by its association with reporter metabolite results.

To go further, we employed the STRING network to examine 371 highly correlated and DEG-significant genes, which were segregated into two distinct clusters: one related to immune system processes and another related to cell division (Appendix A). We used the MLC clustering as default parameters and identified several clusters:

Cluster 1: Focused on cell cycling processes, including genes such as *CDK1* and *CDK4*.

Cluster 2: Centered on immune system processes with genes like *ALOX5*, *S100A8*, *S100A9*, *NCF2*, and *NCF4*, known for their roles in leukocyte arachidonic acid trafficking, phagocyte migration, and NADPH-oxidase activity. 

Cluster 3: Associated with antigen assembly with MHC class II, including *TGFBR2*, *CD4*, and various toll-like and HLA family members. 

Cluster 4: Related to cortical cytoskeleton organisation, featuring *MAP3K1* and *RAC2*.

Interestingly, there were interconnected genes across these clusters, notably *MAP3K1*, *RAC2*, *TGFBR2*, *HMGCR*, and *EZH2*, highlighting a complex interplay of cellular processes in GBM. Targeting these genes may interfere with tumour and microenvironment hypothetically. Additionally, FDA-approved drug targets among these genes revealed 30 such targets, with *CDK4* and *FCGR1A* identified as glioma-specific and *ADORA3* as brain-specific, according to the Human Protein Atlas.

The STRING network analysis of highly positively correlated genes emphasised the involvement of zinc finger family genes connected in the central node, *TREM28* (Appendix A). Several genes from this network belong to the KRAB-ZFPs family. KRAB-ZFPs and *TRIM28* play important roles in regulating stem cell identity, influencing cancer stem cells, and contributing to gene expression networks in the brain [76,77]. SYK is the central node in the negatively correlated gene network (Appendix A), and also SYK in the top 20 high centrality genes in module 67 shows extensive connectivity. SYK is a key signalling molecule in immune responses, playing roles in B and T cell activation, Fc receptor signalling, and the regulation of inflammation that may influence the tumour microenvironment and immune checkpoints [78]. LILRs regulate immune responses, facilitate immune evasion by tumours, and influence the TME, thereby promoting cancer progression and metastasis [79]. Their roles in both immune regulation and direct interactions with cancer cells make them significant players in cancer biology.

Additional genes of interest included *CCDC80* (upregulated in both High and Low TPS), *FOLR2* (negatively correlated), *OGN* (upregulated in High TPS vs. Low TPS), *BCAN* (positively correlated), and *IL12A* (medium positively correlated), each presenting unique insights into the molecular dynamics of GBM. This comprehensive analysis not only highlights potential therapeutic targets but also enhances our understanding of the genetic and molecular landscape of GBM, facilitating targeted treatment strategies. 

## 4. Discussion

This study analysed Glioblastoma (GBM) using integrated network-based approaches. The analysis identified differentially expressed genes (DEGs) and TPS-correlated genes across various tumour purity score (TPS) groups, revealing distinct expression patterns linked to tumour and non-cancerous cell dominance. High TPS was associated with genes such as *SOX11* and *GSX1*, which may promote tumour proliferation, whereas Low TPS was enriched with immune-related genes like *LILRB4*, highlighting the complex interactions within the tumour microenvironment. These variations in TPS were mirrored in GEMs, particularly in the Kynurenine and NAD pathways, which are sensitive in GBM models. Reporter metabolite analysis further identified key metabolites, including arachidonic acid and kynurenine components, emphasising their role in shaping the immune landscape of GBM. Co-regulatory behaviour combined with other results, including TPS correlation, highlight hub genes like *NCAM1*. Together, these findings reveal significant contributions to GBM progression and identify potential therapeutic targets. Building upon this overview of findings, the subsequent sections discuss the specific results in detail.

Several genes are significantly altered within these significantly altered biological processes that may yield results that support the literature. Among them, *BIRC5*, known as Survivin, stands out as a well-recognised anti-apoptotic protein frequently overexpressed in GBM [80]. Survivin’s primary role includes cell death and promoting cell proliferation. Furthermore, *CDK1* and *CDK2*, critical regulators of the cell cycle, play a prominent role in cancer [81]. Dysregulation of these kinases could lead to uncontrolled cell division, a hallmark of GBM. *TP53* is another key gene that is commonly dysregulated in GBM, resulting in the loss of its tumour-suppressing functions [82]. *PTTG1* overexpression is associated with GBM and contributes to the intricate regulation of the cell cycle and mitosis [83].

The pathways identified in the Low TPS results, typically involved in pathogen defence, are manipulated by GBM cells to promote tumour growth and evade immune surveillance. Key tumour–non-cancerous cell interactions include the PD-L1/PD-1 (*CD274*/*PDCD1*) [84] and Galectin-9/TIM-3 (*LGALS9*/*HAVCR2*) [85] pathways, which are significantly upregulated in Low TPS and show a negative correlation with TPS, indicating the influence of the tumour microenvironment (TME). These interactions suppress T-cell activity and promote immune evasion, with PD-L1 binding to PD-1 diminishing T-cell function and Galectin-9 engaging TIM-3, leading to T-cell exhaustion or death. This dual manipulation of immune pathways may create chronic inflammation, support tumour survival, and complicate treatment, but it also highlights novel therapeutic targets to disrupt these mechanisms and improve GBM therapies.

The identified metabolic discrepancies between the NT and GBM models (Low TPS, High TPS, and TP) offered insights into the underlying molecular mechanisms of GBM. Dysregulation of lipid metabolism, linoleate metabolism, and glycosphingolipid biosynthesis could affect cell membrane composition, signalling, and energy utilisation, which are relevant to the GBM mechanism [86]. 

Hepoxilin A3, a lipid mediator derived from arachidonic acid, plays a critical role in the modulation of inflammation and leukocyte functioning [87]. This metabolite is part of eicosanoids, including leukotrienes (Appendix A), which are pivotal mediators in inflammatory responses. Hepoxilin A3 operates within the complex interplay of lipid mediators that orchestrate immune cell behaviour, guiding leukocytes to sites of inflammation via chemotactic signals. Its relationship with oxidative stress is also notable, as it is formed through lipoxygenase pathways that are closely linked to cellular redox states [87]. 

Enzymes from arachidonic acid metabolism, such as *ALOX5*, *ALOX15B*, and *ALOX5AP*, are upregulated in Low TPS and negatively correlate with TPS. This indicates their transcript is influenced by the TME. Arachidonic acid metabolism via ALOX5 codded enzyme not only produces hepoxilin A3 but also generates various hydroperoxy eicosatetraenoic acids (HPETEs) and hydroxy eicosatetraenoic acids (HETEs), such as 15(S)-HPETE, 5(S)-HPETE, 12(S)-HPETE, and 5(S)-HETE, which significantly impact oxidative stress and inflammation by forming lipid peroxides and modulating inflammatory pathways. Abnormal expression of *ALOX5* has been observed in various human cancers, including pancreas, prostate, and colon cancers. *ALOX5* and its by-product metabolites, such as 5-HETE and 5-oxo-ETE, promote tumour cell proliferation that may cause a pro-malignancy path [88,89,90,91]. Metabolites such as 5-HETE, 5-oxo-ETE, 5(S),15(S)-dihydroxyeicosatetraenoic acid (5(S),15(S)-diHETE), and 5-oxo-15(S)-HETE function as hormone-like autocrine and paracrine signalling agents, which may enhance inflammatory and allergic responses, as seen in cancer [92]. These metabolites stimulate eosinophils, playing a crucial role in allergic reactions, and are implicated in inflammation and cancer cell growth. The transformation of heme into biliverdin and bilirubin by heme oxygenase (*HMOX1*) links these pathways. Bilirubin serves as an antioxidant, providing a natural defence mechanism against oxidative damage induced by reactive oxygen species (ROS) and free iron (Fe2+). *HMOX1*, encoded by a rate-limiting enzyme, involved in heme degradation, is significantly upregulated in Low TPS and negatively correlates with TPS, indicating that hypoxic TME dominated *HMOX1* expression in GBM. *HMOX1* helps in tumour cell proliferation and resistance to cell death [93]. *HMOX1* has been demonstrated to promote tumour progression and metastasis in multiple cancers, such as glioma, colorectal cancer, melanoma, and breast cancer [94,95]. Elevated heme degradation may indicate increased oxidative stress and altered iron metabolism [96], which influence cancer progression.

Glycosaminoglycans such as chondroitin sulfate and keratan sulfate are structural components of the extracellular matrix, which undergo degradation and contribute to cellular signalling pathways that modulate inflammation and oxidative stress. These interactions exemplify the intricate network of biochemical pathways that manage and connect oxidative stress, inflammation, immune response, and metabolic health, underscoring the importance of understanding these relationships in the context of diseases characterised by inflammation and oxidative stress.

*ENPP1*, coded a gene in Vitamin B2 metabolism, is responsible for the production of *FMN*, which shows no significant change according to tour results. However, the substrates of this reaction are used in riboflavin metabolism. *ACP5* and *ACP6*, which are involved in riboflavin production, are favourable genes based on survival results. A study suggests that high riboflavin intake may reduce the risk of glioma [97]. In a study with a model organism, researchers have explored the role of amphioxus (a cephalochordate) proteins ACP3 and ACP5, which contain the apextrin C-terminal (ApeC), in antimicrobial immune responses. They demonstrated that both proteins could bind and aggregate microbes. *ACP3* specifically regulates an intracellular pathway that involves *TRAF6* and *NF-κB,* indicating potential analogous in human immune systems in terms of functionality [98]. *ACP3* is a prostatic acid phosphatase that also regulates prostate cancer cell growth by dephosphorylating *ERBB2*, which is a part of the adjacent network of our target gene *CHST2* in our previous study [99] that shows a slight negative correlation with TPS and deactivates MAPK-mediated signalling [100]. *CYP1B1* is a critical metabolic enzyme for melatonin metabolism, and its role in cancer has been emphasised; this suggests that dysregulated expression of *CYP1B1* is associated with the tumour’s clinical stage, grade, survival, and immune microenvironment [101]. On the other hand, the gene responsible for producing FAD, *FLAD1*, appears to be unfavourable. Lastly, differences in peptide metabolism among models might relate to immune system alterations in GBM. Consequently, peptides that are integral to amino acid metabolism and immune responses could influence the tumour microenvironment. Investigating the connections between these metabolic alterations in GBM could provide valuable insights into the disease mechanisms and potentially lead to the development of novel therapeutic strategies.

The biological relevance to the GBM mechanism becomes evident when considering that these URGA metabolites are associated with several critical metabolic processes that are relevant to cancer cell growth, invasion, and survival. The dysregulation of these processes may contribute to the development and progression of GBM. Downregulated metabolites are often associated with processes that are critical for cancer cell survival and proliferation. GBM cells seem to exhibit altered amino acid metabolism, potentially relying on alternative sources of energy like ketones. Moreover, the dysregulation of lipids involved in signalling pathways suggests that GBM cells may manipulate signalling cascades to promote their growth and evade mechanisms that would normally limit their proliferation or induce cell death.

The single gene deletion analysis, by utilising GEMs, identified key metabolic genes specific to GBM models, such as *KYNU*, *GUSB*, *SLC27A5*, *DCK*, *CMPK1*, and *NADSYN1*, emphasising their roles in essential metabolic pathways (Figure 5 and Appendix A). *KYNU* and *GUSB* come to the fore in terms of their expression profile. Particularly, *KYNU*, a critical enzyme in the kynurenine and NAD synthesis pathways via tryptophan, showed a strong negative correlation with TPS, identifying it as a sensitive target involved in immune modulation and cancer metabolism.

However, *KYNU* and kynurenine metabolite through the tryptophan pathway are closely tied to immune regulation through the kynurenine pathway, activating T-lymphocytes and suppressing the innate immune system [49]. Targeting *KYNU* could interfere with NAD synthesis, potentially disrupting both cancer and immune cell functions, thereby inducing unintended immunosuppressive effects. Thus, while the NAD pathway presents therapeutic opportunities, inhibiting *KYNU* could compromise normal immune responses, complicating treatment strategies. At the same time, this situation could be beneficial when treating a patient who undergoes chemotherapy. This underscores the importance of differentiating between tumour-specific metabolic targets and broader immune functions in therapy development [50].

By utilising an integrated approach that combines DEG analysis, correlation and co-expression analysis, and tissue-specific expression information, this study identified novel key genes, such as *GSX1*, *SOX11*, *LILRB4*, and *NCAM1*, which may be crucial players in the GBM mechanism making them promising targets for therapeutic strategies. Our findings demonstrate the potential of this approach to evaluate and manipulate these target genes. By integrating tumour purity score (TPS) into the gene expression profile, we proposed a dual therapeutic strategy: one focusing on TME interactions, particularly immune response-related components such as *LILRB4*, and the other targeting tumour intrinsic cellular mechanisms influenced by gene expression perspective, including transcription factors, such as *GSX1* and *SOX11*. Additionally, gene-to-gene co-regulation and co-expression perspective *NCAM1* featured as a promising central node. Additionally, the use of TPS group-specific GEM analysis, consequently, identifying reporter metabolites and essential genes, has revealed altered mechanisms and highlighted the effects of sample heterogeneity on gene expression. This enhances our ability to trace the origins of transcript sources within the sample that navigates targeted interventions. 

In summary, this study emphasised that the heterogeneity of samples could affect statistics and may also navigate tissue-specific therapeutic intervention. This research could provide crucial insights into GBM’s complex mechanisms, paving the way for targeted therapies that enhance treatment precision and effectiveness.

This study utilised bulk RNA-Seq data to explore the transcriptomic landscape of glioblastoma, providing substantial insights. However, this approach has inherent limitations. Firstly, bulk RNA-Seq can hide cellular heterogeneity, which may mask the cell-to-cell interaction information, resulting in an averaged gene expression profile that may neglect rare but clinically significant cell types. Although TPS-based analysis is employed to differentiate transcript sources, it cannot identify specific cell types or their roles. Moreover, the loss of spatial context in gene expression data handicaps the understanding of cellular interactions within the tumour microenvironment. Critical interactions, such as paracrine and juxtacrine signalling, which involve local signal exchanges, are crucial for understanding cell behaviour in tissue context, particularly in GBM, due to their invasive nature.

The integration of single-cell RNA sequencing (scRNA-Seq) and spatial transcriptomics offers a promising enhancement. scRNA-Seq could identify distinct cellular populations within glioblastoma tumours, providing deeper insights into tumour heterogeneity that bulk RNA-Seq may not achieve. This approach may clarify the roles of specific cell types in tumour progression. Additionally, spatial transcriptomics maps the cell populations to their original locations within the tumour, adding a spatial dimension that enriches our understanding of cellular architecture and microenvironment interactions.

While the scRNA-Seq process includes cell isolation, which might disrupt natural cell-to-cell interactions, combining it with spatial transcriptomics and bulk RNA-Seq helps maintain the broader biological context and offers improved resolution. Integrating these advanced methodologies enables more detailed exploration of the tumour microenvironment, potentially revealing new biomarkers and therapeutic targets.

In conclusion, this study demonstrates that an integrated network-based approach is an effective strategy for unravelling the complex molecular landscape of GBM. By leveraging differential gene expression (DEG) analysis, tumour purity score (TPS) correlation, co-expression network analysis, and genome-scale metabolic modelling (GEMs), we gained detailed insights into the interactions between gene expression, metabolism, and the tumour microenvironment. This comprehensive approach not only identifies actionable therapeutic targets but also evaluates their suitability, considering potential risk in terms of cell fate. The findings provide a foundation for developing more precise and tissue-oriented therapeutic strategies for GBM, enhancing the potential for targeted interventions that address the intricacies of the disease.

## Figures and Tables

**Figure 1 biomedicines-12-02237-f001:**
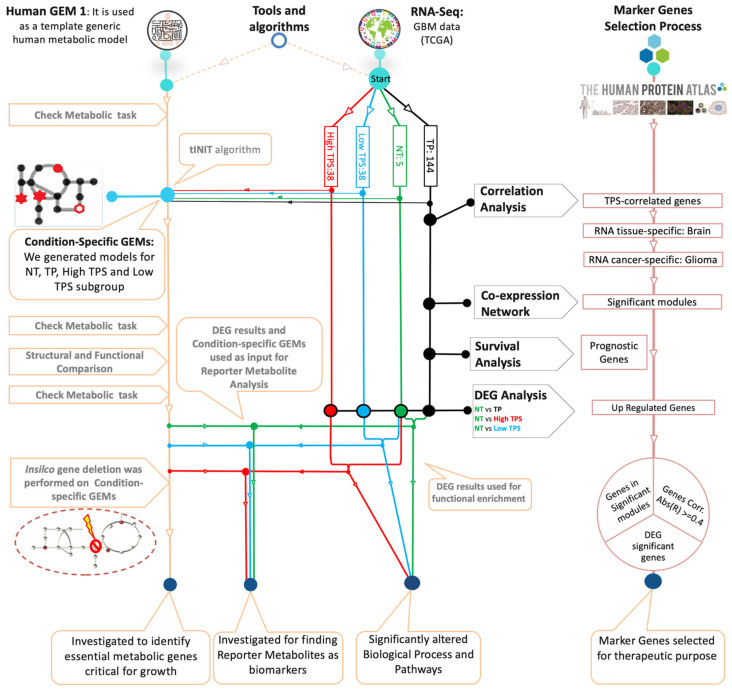
Study overview for the analysis of Glioblastoma Multiforme. The diagram illustrates the integration of various analytical approaches to explore the disease mechanisms of GBM. Patient-specific RNA-Seq data and the Human GEM 1 were used to generate condition-specific GEMs using the tINIT algorithm. Key analytical steps include co-expression network analysis, survival analysis, correlation analysis, DEG analysis, functional enrichment analysis, reporter metabolite analysis, and gene essentiality analysis. These analyses help identify potential biomarkers and drug candidates and understand the mechanism of the disease. Each node represents a specific analysis, connected by lines indicating the flow. Normal tissue (NT), primary tumour (TP), subgroup with high tumour purity score (High TPS), and subgroup with low tumour purity score (Low TPS).

**Figure 2 biomedicines-12-02237-f002:**
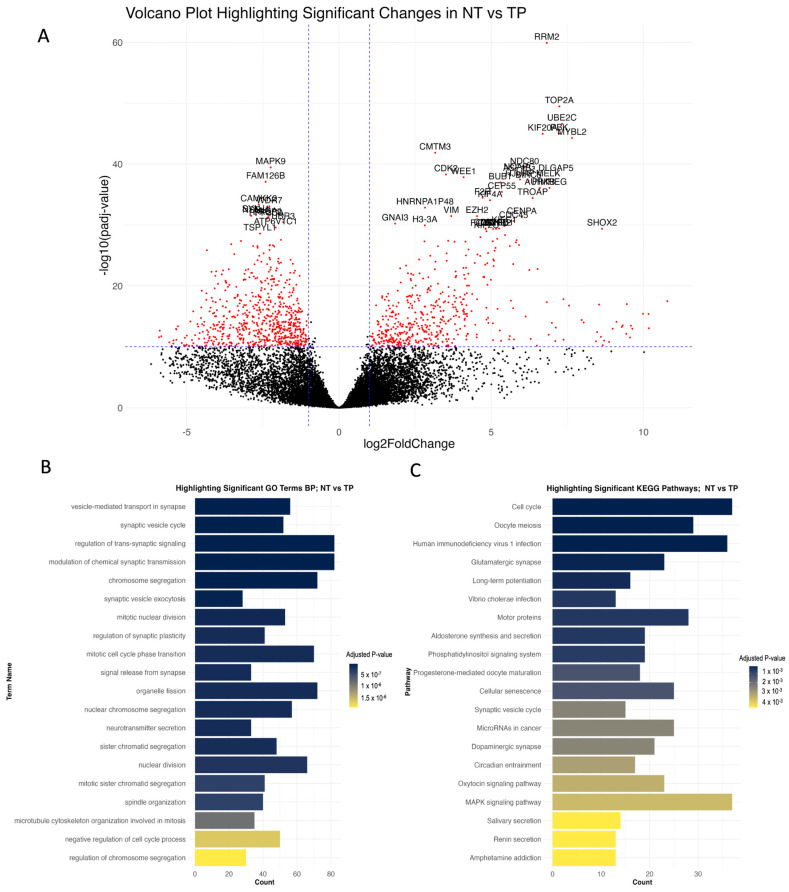
Comparative analysis of gene expression and gene-set enrichment analysis in NT vs. TP conditions. (**A**): Volcano plot displaying comparative analysis results between NT and TP conditions. Red dots indicate genes that are significantly altered. (**B**): Bar chart detailing the top significantly altered biological processes in TP relative to NT, with the x-axis representing the number of significantly altered genes in each process. (**C**): Bar chart showing the top significantly altered KEGG pathways in TP compared with NT.

**Figure 3 biomedicines-12-02237-f003:**
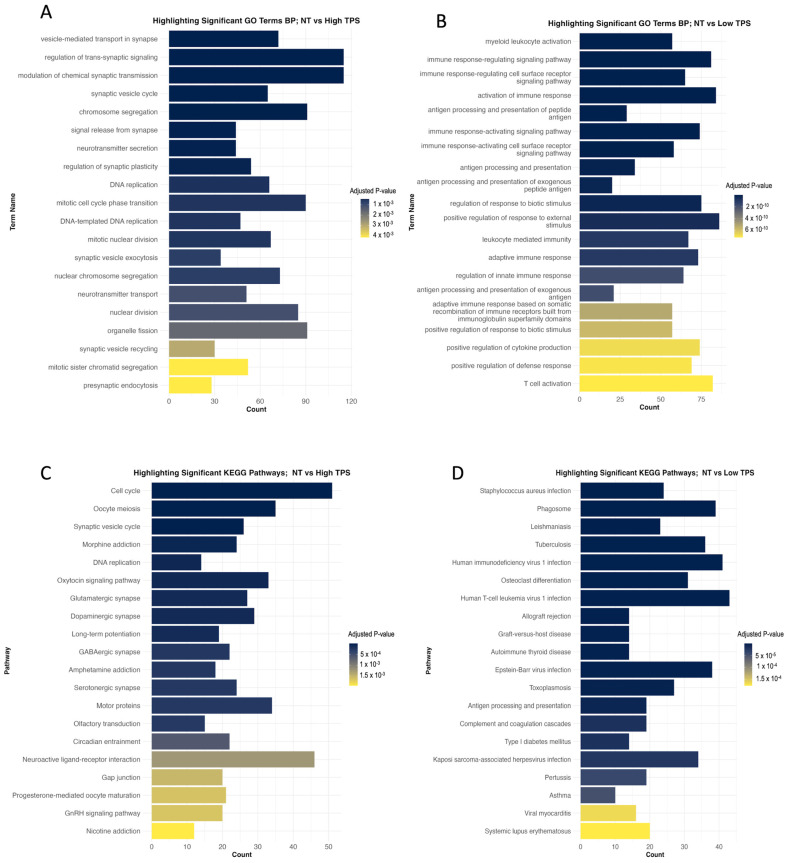
This figure presents the results of gene-set enrichment analyses in TPS contexts. (**A**): The bar chart displays the top biological processes significantly altered in the High TPS subgroup, which may indicate highly proliferative tumour cell behaviour. (**B**): The bar chart shows the top biological processes significantly altered in the Low TPS subgroup, which may indicate immune responses. (**C**): The bar chart illustrates the top KEGG pathways that are significantly altered in High TPS. (**D**): The bar chart shows the top KEGG pathways significantly altered in Low TPS. Each panel systematically categorises the top 20 affected processes and pathways, providing insights into how gene expression varies with changes in tumour purity, which is useful for understanding tumour behaviour and identifying potential therapeutic targets.

**Figure 4 biomedicines-12-02237-f004:**
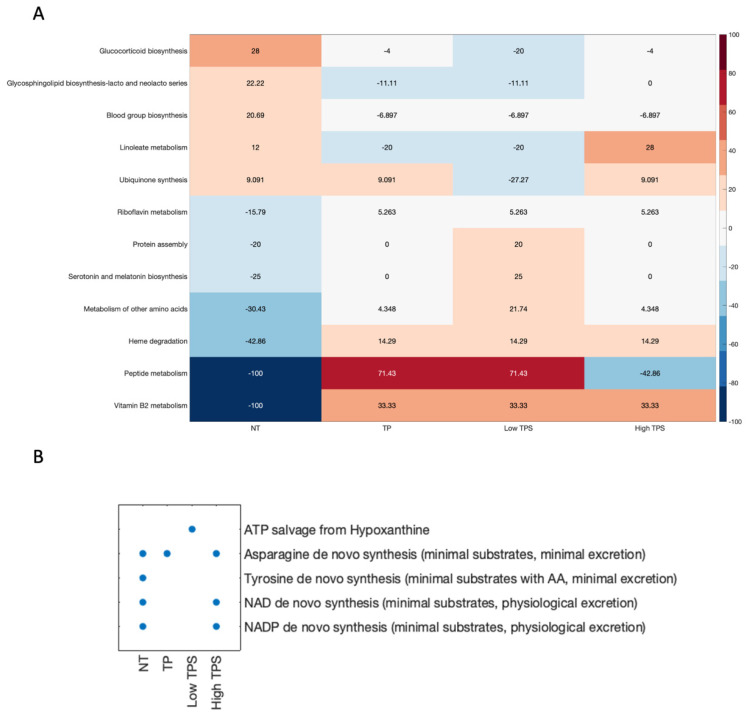
Metabolic task performance and structural comparison of metabolic models. (**A**): Heatmap illustrates the structural comparison of metabolic models. Each cell represents the deviation of sub-coverage from the average, with positive values indicating higher than average sub-coverage and negative values vice-versa. The colour gradient from red to blue corresponds to the degree of deviation. (**B**): Scatter plot categorising the “pass” or “fail” of metabolic tasks in models. Each dot represents a “pass” of corresponding metabolic tasks, emphasising the functional capabilities under different tumour purity effects.

**Figure 5 biomedicines-12-02237-f005:**
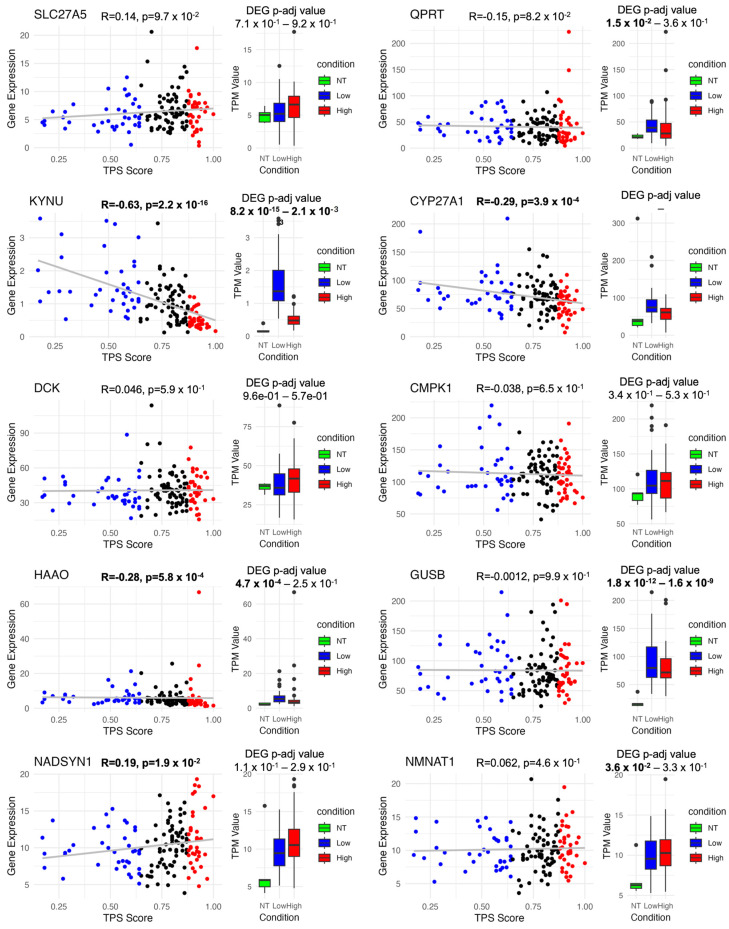
Correlation and differential expression of 10 essential genes for cancer models. This figure illustrates that although many essential genes are common across various models, 10 genes are uniquely crucial to cancer models. The figure features a scatter plot correlation tumour purity score (TPS) with gene expression and a box plot comparing gene expression across NT vs. Low TPS and High TPS conditions. Blue (Low TPS), black (Mid TPS), and red (High TPS). The grey line represents the linear regression fit (R).

**Figure 6 biomedicines-12-02237-f006:**
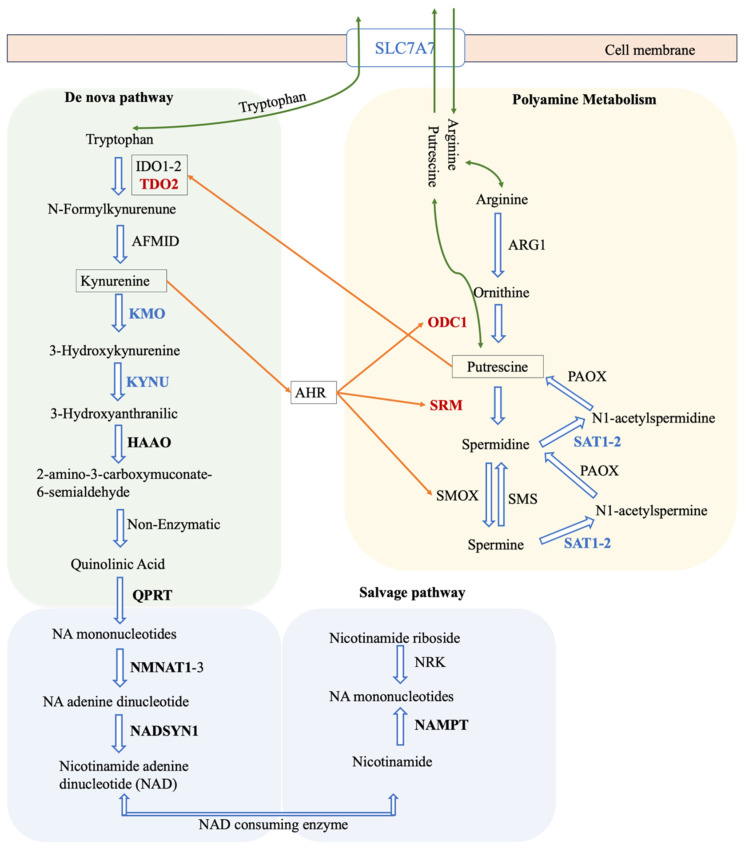
NAD+ biosynthesis via tryptophan de nova biosynthesis and integration with polyamines. This diagram outlines the kynurenine and polyamine metabolism with the NAD+ salvage pathway, underscoring their significant roles in immune modulation and cancer. It shows the enzymatic conversion of tryptophan and arginine into crucial intermediates impacting immune responses and cellular health, along with the essential recycling of nicotinamide into NAD+. Connections through shared enzymes, *SLC7A7* and *AHR*, emphasise their collective influence on immune function and cancer pathways.

**Figure 7 biomedicines-12-02237-f007:**
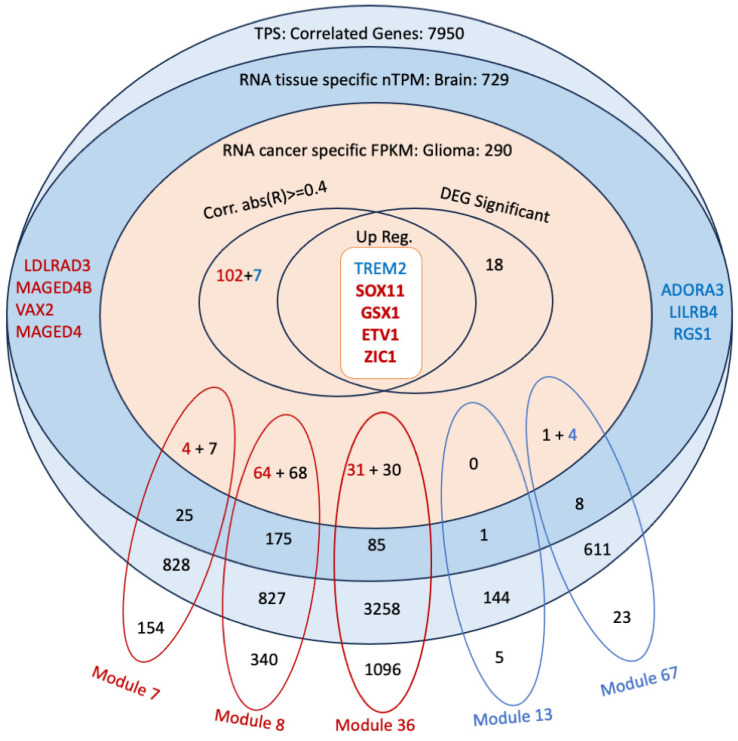
Candidate gene selection framework for GBM therapy. This figure illustrates a multifaceted approach to identifying potential therapeutic targets for GBM, integrating several analytic techniques. The Venn diagram merges DEG analysis, co-expression network analysis, and correlation analysis along with tissue expression information from the Human Protein Atlas, which includes the expression pattern of a gene across tissues. The colour-coding distinguishes between genes: blue for negatively correlated genes and red for positively correlated genes. The labelled modules, colour-coded to indicate significant matches with either positively or negatively correlated genes, represent clusters that consist of genes from each analytical layer. This comprehensive selection process aims to distinguish genes central to the pathophysiology of GBM, which could be key targets for developing more precise therapeutic strategies.

**Figure 8 biomedicines-12-02237-f008:**
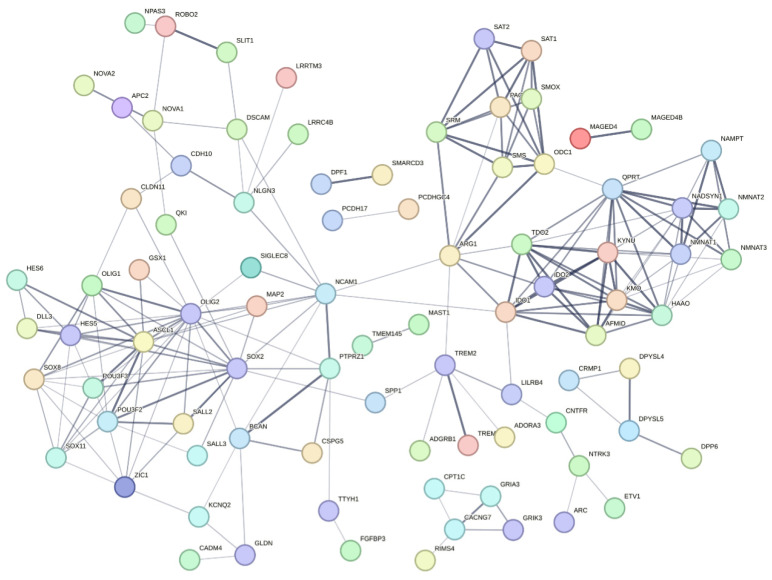
STRING network of correlated and significant genes in GBM. To identify therapeutic targets for GBM, we employed a comprehensive analytical approach that integrates DEG analysis, correlation analysis, and tissue expression data from the Human Protein Atlas. We began by selecting genes with correlation coefficients (absolute value of R ≥ 0.4) from Figure 7. Additionally, we included upregulated DEGs from brain-expressed genes depicted in Figure 7 and incorporated relevant genes from Figure 6. These genes were then analysed using the STRING database, with unconnected genes hidden and default parameters applied. This extensive analysis highlighted critical marker genes that play significant roles in the dynamics of GBM tumours. The network genes are stored in Appendix A.

**Figure 9 biomedicines-12-02237-f009:**
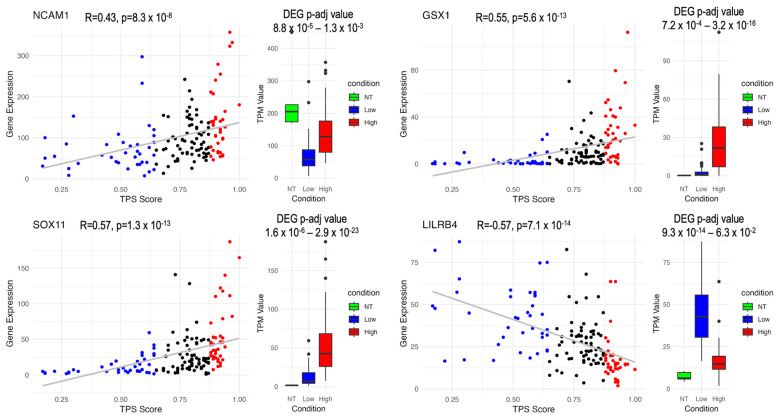
Differential expression and correlation profiles of marker genes in GBM according to TPS: This figure presents the DEG and correlation profiles of *GSX1*, *SOX11*, *LILRB4*, and *NCAM1* based on TPS (Tumor Proportion Score). *GSX1*, *SOX11*, and *NCAM1* are positively correlated with TPS, while *LILRB4* is negatively correlated. DEG *p*-adj value represents the comparison of Low TPS to NT and High TPS to NT, respectively. Categories include NT (Normal Tissue), Low (Low TPS group), and High (High TPS group). Blue (Low TPS), black (Mid TPS), and red (High TPS). The grey line represents the linear regression fit (R).

**Table 1 biomedicines-12-02237-t001:** Upregulated gene-related metabolites. This table lists metabolites associated with metabolic genes that are upregulated, offering a basic exploration of their roles in various biological processes.

Upregulated Gene Affected Metabolites	Short Summary
Lipid Metabolism and Cholesterol-Related Metabolites	
13(S)-HPODE	Associated with breast cancer cell proliferation and invasion.
5(S)-HPETE	Involved in the arachidonic acid pathway.
Cholesterol-ester	Implicated in steroid hormone biosynthesis and lipid metabolism.
Dehydrocholic acid	Investigated for potential anticancer properties.
Taurodeoxycholate and lithocholate	Involved in cholesterol metabolism.
Nucleotide Synthesis and Purine Metabolism	
GAR (Glycinamide ribonucleotide)	Involved in purine biosynthesis and can affect cancer cell growth.
Adenine, threonine, tetrahydrofolate	Critical for nucleotide synthesis and DNA methylation.
Extracellular Matrix and Invasion	
Hyaluronate	A component of the extracellular matrix plays a key role in cancer cell invasion and metastasis.
Pentose Phosphate Pathway and Glycogen Metabolism	
Ribose-5-phosphate	A component of the pentose phosphate pathway, involved in nucleotide synthesis, impacts cancer cell proliferation.
Glucose-1-phosphate	Involved in glycogen, energy, and glycosylation metabolism.
Glycosphingolipid Metabolism	
Galactosylglycerol	Involved in glycosphingolipid metabolism, cell signalling, and adhesion.
Inflammatory Response and Cell Survival	
Hepoxilin A3	Implicated in inflammatory responses, chemoattractant for neutrophils.

**Table 2 biomedicines-12-02237-t002:** Downregulated gene-related metabolites. This table lists metabolites associated with metabolic genes that are downregulated, offering a basic exploration of their roles in various biological processes.

Downregulated Gene Affected Metabolites	Short Summary
Amino Acid Metabolism	
Glutamate and Glutamine	Key players in brain metabolism, including neurotransmission, potentially contribute to the disease pathophysiology and might indicate altered energy and biosynthetic needs in GBM cells [47].
Phenylalanine and its derivatives Phenylpyruvate and Hydroxyphenylpyruvate	It may be an indicator of GBM’s metabolic adaptations, affecting cancer progression. Glioma cell consumes more Phenylalanine, a precursor for neurotransmitters. Hydroxyphenylpyruvate influences oxidative stress via tyrosine metabolism [48].
Tyrosine	It plays a role in multiple signalling pathways. Its involvement could influence cell signalling, potentially contributing to disease development. Tyrosine de novo synthesis metabolic task defective in all GBM models (Figure 4B) [49,50,51].
Ketone Metabolism	
Acetoacetate	An alternative energy source via ketone metabolism that may indicate the energy demands of rapidly proliferating cells [52].
Lipid Signaling	
Phosphatidylinositol (PI)	Phosphatidylinositol (PI) and its derivatives, including phosphatidylinositol-4,5-bisphosphate (PIP2) and phosphatidylinositol-3,4,5-trisphosphate (PIP3), are involved in signalling cascades linked to cell growth and survival. Dysregulation of them may be indicative of potential disruptions in crucial signalling pathways, contributing to tumour growth and evasion of cell death [53,54].

## Data Availability

The RNA-seq data used in our research is publicly available and can be accessed through the GDC portal under Project ID: TCGA-GBM at the following URL: https://portal.gdc.cancer.gov/ (accessed on 2 March 2023). The code is available upon request.

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
