# Peer review of "Unveiling the Molecular Mechanisms of Glioblastoma through an Integrated Network-Based Approach"

_biomedicines, 2024, doi:10.3390/biomedicines12102237_

Round 1

Reviewer 1 Report

Comments and Suggestions for Authors

The authors present an original paper of a computational background, aiming to assess glioblastoma related transcriptomes from The Cancer Genome Atlas. Several models and networks were used in this assessment, reaching an analysis for some genes and metabolites, in addition to some molecular mechanisms driving glioblastoma.

I believe that this manuscript is well prepared, yet holds some points that can be improved:

1-      The objectives of the study are not clearly presented. Objectives should be made straight forward.

2-      The legend of Figure 1 needs to be more elaborated and clarified, as it is still complicated to understand.

3-      Some parts of the results section better fit the discussion, especially when explaining the results and comparing them with other studies.

4-      The discussion would benefit from a small recap of the results in the first paragraph.

5-      I suggest adding some future perspectives, and how could other research build on these findings.

Author Response

Comments and Suggestions for Authors

The authors present an original paper of a computational background, aiming to assess glioblastoma related transcriptomes from The Cancer Genome Atlas. Several models and networks were used in this assessment, reaching an analysis for some genes and metabolites, in addition to some molecular mechanisms driving glioblastoma.

Response: We sincerely thank you for the constructive comments and valuable suggestions, which have greatly contributed to the improvement of our manuscript. We have carefully addressed each point raised, and our detailed responses are provided below:

I believe that this manuscript is well prepared, yet holds some points that can be improved:

Comments 1-      The objectives of the study are not clearly presented. Objectives should be made straight forward.

Response 1: We appreciate the feedback. The objectives of the study have been re-examined and modified to ensure it is clearly stated and directly aligned with the overall aims of the research.

Comments 2-      The legend of Figure 1 needs to be more elaborated and clarified, as it is still complicated to understand.
Response 2: Thank you for highlighting this point. We have updated Figure 1 to provide clearer and more detailed explanations, making the figure easier to understand.

Comments 3-      Some parts of the results section better fit the discussion, especially when explaining the results and comparing them with other studies.
Response 3: Thank you for this suggestion. We have carefully reviewed the results section and relocated the relevant parts to the discussion. The sections have been modified to better integrate the results with comparative insights from existing literature.

Comments 4-      The discussion would benefit from a small recap of the results in the first paragraph.

Response 4: Thank you for this valuable suggestion. We have revised the first paragraph of the discussion to include a brief recap of the key results, enhancing the flow and coherence of the discussion.

Comments 5-      I suggest adding some future perspectives, and how could other research build on these findings.

Response 5: We appreciate this suggestion. We have added a section discussing future perspectives, highlighting how our findings could guide subsequent research.

Reviewer 2 Report

Comments and Suggestions for Authors

Dear Authors, 

Congratulations on the manuscript. The following are some suggestions for improving its quality.

1)The scientific merit is remarkable however we need to include in the introduction which is quite substantial some guidance on complex diagnosis through imaging, and treatment, also referring to the WHO 2021 classification, so please read and cite these two articles:

Advancements in Glioma Care: Focus on Emerging Neurosurgical Techniques. Biomedicines. 2024;12(1):8. https://doi.org/10.3390/biomedicines12010008 

Clustering Functional Magnetic Resonance Imaging Time Series in Glioblastoma Characterization: A Review of the Evolution, Applications, and Potentials. Brain Sci. 2024, 14, 296. https://doi.org/10.3390/brainsci14030296

2) From line 87 to 104 a more concise elaboration should be made

3) table 2 should have references 

4) Figure 5, Statistically significant p-values should be indicated in bold.

5)Two additional paragraphs should be made after the discussion; limitations and conclusions.

Once again congratulations on the work done I look forward to reading the new manuscript with all my directions incorporated.

Comments on the Quality of English Language

In some sections the manuscript should be made more impersonal, there are too many “we”

Author Response

Comments and Suggestions for Authors

Dear Authors, 

Congratulations on the manuscript. The following are some suggestions for improving its quality.

Response: We are grateful to you for the feedback and insightful suggestions. Your comments have significantly contributed to enhancing the quality of our manuscript. Please find our detailed responses below:

Comments 1)The scientific merit is remarkable however we need to include in the introduction which is quite substantial some guidance on complex diagnosis through imaging, and treatment, also referring to the WHO 2021 classification, so please read and cite these two articles:

Advancements in Glioma Care: Focus on Emerging Neurosurgical Techniques. Biomedicines. 2024;12(1):8. https://doi.org/10.3390/biomedicines12010008 

Clustering Functional Magnetic Resonance Imaging Time Series in Glioblastoma Characterization: A Review of the Evolution, Applications, and Potentials. Brain Sci. 2024, 14, 296. https://doi.org/10.3390/brainsci14030296

Response 1: Thank you for suggesting these relevant references. We have reviewed the recommended articles and integrated the suitable information into the introduction. The suggested citations have been added to align the manuscript with the latest advancements in glioma care and diagnosis.

Comments 2) From line 87 to 104 a more concise elaboration should be made

Response 2: We appreciate your feedback. This section has been re-evaluated and rewritten to present a more concise and focused elaboration, particularly refining the purpose and scope of the study.

Comments 3) table 2 should have references 

Response 3: Thank you for pointing this out. We have added the relevant references to Table 2.

Comments 4) Figure 5, Statistically significant p-values should be indicated in bold.
Response 4: Thank you for your meticulous observation. We have modified Figure 5 to highlight statistically significant p-values in bold, enhancing the clarity of the figure.

Comments 5)Two additional paragraphs should be made after the discussion; limitations and conclusions.

Response 5: Thank you for this recommendation. We have added separate paragraphs outlining the limitations of the study and the main conclusions, emphasizing the scope for future research and the practical implications of our findings.

Once again congratulations on the work done I look forward to reading the new manuscript with all my directions incorporated.

Comments on the Quality of English Language

In some sections the manuscript should be made more impersonal, there are too many “we”

Response: Thank you for this linguistic feedback. We have revised the manuscript to adopt a more impersonal and formal tone, reducing the usage of "we" to enhance the academic presentation of the text.

Round 2

Reviewer 2 Report

Comments and Suggestions for Authors

Dear authors congratulations,

Regarding the surgical treatment mentioned in the introduction could you also add the reference to skull base surgery website: https://doi.org/10.3390/jcm13092712 ; https://doi.org/10.3390/jcm13133701